# Sport Locus of Control and Perceived Stress among College Student-Athletes

**DOI:** 10.3390/ijerph16162823

**Published:** 2019-08-08

**Authors:** Shelley L. Holden, Brooke E. Forester, Henry N. Williford, Erin Reilly

**Affiliations:** 1Department of Health, Kinesiology, and Sport, University of South Alabama, Mobile, AL 36688, USA; 2Department of Kinesiology, University of Auburn at Montgomery, Montgomery, AL 36117, USA

**Keywords:** stress, locus of control, athletes

## Abstract

The purpose of the study was to analyze athletes’ motivation for sport participation as it related to their locus of control. Research was conducted at two Division I universities in the southeastern United States. Participants were given the Sport Locus of Control and Perceived Stress among College Athletes surveys. There were 126 participants with a mean age of 19.69 ± 1.32 years. A Pearson correlation (r) was performed to determine a significant relationship between perceived stress and locus of control. Results indicated a significant negative relationship between the two variables (r = −0.393 and *p* = 0.001) (a moderate relationship). As perceived stress scores increased, locus of control scores decreased. Correlations related to perceived stress were gender (r = 0.323, *p* = 0.000), and grade point average (GPA) (r = −0.213, *p* = 0.01). The only other independent variable that was significantly related to locus of control was being on an academic scholarship (r = −0.203, *p* = 0.025). Athletes who have an external locus of control feel that they have little control over their circumstances. Findings of this study give coaches another factor to consider in retaining and getting the most from their athletes.

## 1. Introduction

The number of student-athletes participating in the National Collegiate Athletic Association (NCAA) is at an all-time high with approximately 491,930 student-athletes competing in sports at the NCAA championships during the 2016–2017 academic school year [1]. Paralleling the rise of NCAA athletic participation is the rise of stress among college students, including student-athletes. About 30 percent of the 195,000 student respondents to a recent American College Health Association (ACHA) survey reported having felt depressed in the last 12 months and 50 percent reported having felt overwhelming anxiety during the same period [2]. A primary concern regarding the prevalence of mental illness among student-athletes is it may affect their success in academics and athletics as well as their general well-being [2]. Depression and anxiety, a byproduct of stress, have been found to be significant predictors of a lower grade-point average and poor athletic performance and they also seem to be highly correlated with suicide [2].

### 1.1. Perceived Stress

Stress, injury, and over-training syndrome are common in elite sports [3]. Stressors of student-athletes may come in many forms such as playing time, injuries, discontentment with coaching style, poor academic performance, relationships with teammates and their win and loss record. Interestingly, 40 percent of all collegiate student-athletes receiving scholarship money transfer, quit school completely, or do not graduate within six years [4]. Moreover, athlete stress is not limited to the highest level of competition. Adolescent athletes also experience a number of stressors. These include competitions, regular social evaluation and criticism, family and peer influences, and well as academic commitments [5,6,7,8]. Stress can result in the following symptoms: fatigue, hypertension, headaches, depression and anxiety, but stress is not limited to athletes in the athletic setting. Stress is influenced by factors such as major life events, daily hassles, and coping resources, so stress can certainly change over time [9]. Further, Britton et al. examined athletes’ relationship with perceived stress and reactivity [10]. Stress reactivity has been measured using physiological (e.g., heart rate variability, blood pressure, cardiac output) and is defined as a disposition underlying individual differences in physiological and psychological stress responses [11]. The results indicated that adolescent athletes who are highly reactive experience greater levels of stress over time, feeling that their lives are somewhat uncontrollable, and they find it difficult to cope [10]. In other words, more reactive adolescent athletes experience more stress and experience lesser satisfaction across life domains [10]. Therefore, there seems to be a relationship between perceived stress and an individual’s locus of control.

### 1.2. Locus of Control and Self Determination Theory

Locus of control refers to an individual’s beliefs about the extent of the control one has over things that may have happened to them [12]. That is, the source of control in one’s life is believed to be determined by a person’s orientation toward personality dimensions (internal, external, and powerful chance), as conceived by Levenson [13]. Self-determination theory (SDT) is based upon the theory of human motivation, development, and wellness. More specifically, it focuses on the types of motivation (intrinsic versus extrinsic) as predictors of performance and well-being outcomes [14]. However, motivation is not viewed as solely intrinsic (inherent rewards of the behavior itself) or extrinsic (motivated by external rewards such as money or acclaim); rather, it is viewed as a continuum between self-determined and non-self-determined behaviors [14].

Internal locus of control is described as outcomes resulting from one’s actions which can facilitate perceived competence and increase intrinsic motivation. On the contrary, external locus of control is outcomes resulting from external forces (e.g., luck or chance) which tend to undermine intrinsic motivation. Therefore, locus of control may impact an individual ability to handle stressors of all types and it plays a key role in how an individual perceives stressors. Further, research has indicated that internally controlled individuals tend to use time more effectively and constructively to react when faced with obstacles. Athletes participating in team sports reported higher external locus of control [15,16]. Prior research has also determined that neuroticism (anxiety, fear, moodiness, worry, envy, frustration, jealousy, and loneliness) as a personality trait is associated with higher perceived stress, lower perceived control, higher stressor intensity, and using avoidance as a coping strategy [17,18].

Therefore, the purpose of this study was to analyze perceived stress in student-athletes as it related to their locus of control. Instruments administered in this study have been used in prior studies to measure perceived stress in college students and athletes [19,20]. Further, this line of research is in line with SDT and the hierarchical model of intrinsic and extrinsic motivation [14,21].

## 2. Methods

### 2.1. Participants

Research was conducted at two Division I universities in the southeastern United States. Institutional Review Board (IRB) approval to conduct the study was obtained by both institutions. Head coaches from all athletic teams at each institution were contacted requesting their team’s participation in the study. Eight different sports including softball, tennis, volleyball, basketball, men’s and women’s track and field, cheerleading, and soccer participated in this study.

### 2.2. Instrumentation

Student-athletes were given a short demographic survey consisting of items such as age, gender, sport participation, major, race, year in school, grade point average (GPA) and scholarship. After the demographic portion, participants completed the Perceived Stress Scale (PSS) [22] and the Sport Locus of Control (SIES) [23,24] surveys.

The PSS is designed to measure the degree to which situations in one’s life are considered stressful [25]. Further, the PSS is the most widely used psychological instrument for measuring the perception of stress [9]. This survey consists of 10 questions ranked from 0 to 4 dealing with participants’ feelings and thoughts during the last month. In each question, participants are asked how often they felt a certain way. PSS scores are obtained by reversing responses (ex. 0 = 4, 1 = 3, 2 = 2, etc.) to the four positively stated items (4, 5, 7, 8) then summing across all scale items [25].

The SIES is used to assess the relative value of internal and external sources of reinforcement perceived by an athlete [22]. The SIES consists of 31 items with two alternatives for each question. Each question has an external control factor alternative, with a maximal external control score of 23. Participants’ external control score is used to calculate the internal control score by the following formula: Internal Control Score = 23 − External Control Score.

### 2.3. Procedure

Lead researchers first trained secondary researchers on the procedure for administering the instruments. Once all researchers were trained on the administration of the instruments, head coaches from the university athletic teams were contacted requesting permission to survey student-athletes from their respective teams. When consent was received from the head coach, researchers and coaches determined a day and time for the student-athletes to complete the surveys. Coaches were requested to attend the survey session to verify athlete attendance. Once all players were present and the researcher gave the survey instructions, coaches exited the survey room until all surveys were completed. Researchers met with each team individually and were responsible for monitoring the completion of the surveys and demographic questionnaire. At the beginning of each survey session, the researcher explained the purpose and the details of the survey to the student-athletes. Student-athletes were informed of their right not to participate and of the confidentiality of their responses.

## 3. Results

There were 126 participants in this study aged 19.69 ± 1.32 years. Seventy-eight percent (*n* = 98) of the participants were female and 22% male (*n* = 28). There were eight different sports represented including softball (*n* = 21), tennis (*n* = 5), volleyball (*n* = 12), basketball (*n* = 38), track and field (*n* = 19), cheerleading (*n* = 12), and women’s soccer (*n* = 19). Twenty-eight percent of the student-athletes were on academic scholarships and 80% were on an athletic scholarship. The mean GPA in the sample was 3.30 on a 4.0 scale. Other demographic descriptive statistics are included in Table 1.

In order to determine if there was a significant relationship between perceived stress and locus of control, a Pearson correlation (r) was performed. Results indicated there was a significant negative relationship between the two variables (r = −0.393 and *p* = 0.001) (moderate relationship). The statistic indicates that as perceived stress scores increased, locus of control scores decreased. Correlations related to perceived stress were gender (r = 0.323, *p* = 0.000) and GPA (r = −0.213, *p* = 0.01). The only other independent variable that was significantly related to locus of control was being on an academic scholarship (r = −0.203, *p* = 0.025) (see Table 2).

## 4. Discussion

Data analysis in the current study indicated a moderate significant relationship between college student-athletes′ locus of control and perceived stress (Table 2, Figure 1). This finding is what the authors expected and is consistent with prior research findings [26]. That is, student-athletes who have an external locus of control feel that they have little control over their circumstances. If athletes believe they are powerless to change their circumstance, they would perceive more stress than those who feel they are in control of their own destiny. Also, if student-athletes on a scholarship felt they may be in danger of losing their scholarship, it could add to their level of perceived stress. This would be particularly true for those who felt they had little control over their circumstances.

A main distinction in SDT is between controlled and autonomous motivation. Controlled motivation consists of external regulation, where behavior is a function of external contingencies of reward or punishment and introjected regulation, and where the regulation of action has been partially internalized and is energized by factors such as approval, avoidance of shame, contingent self-esteem, and ego-involvements [27]. People who are controlled experience pressure to think, feel, or behave in particular ways. Autonomous motivation is composed of both intrinsic and extrinsic motivation where people have identified with an activity’s value and ideally have integrated it into their sense of self [27]. People who are autonomously motivated experience purpose, or a self-endorsement of their actions. Both controlled and autonomous motivations tend to energize and direct behavior which is in contrast to amotivation (lack of intention and motivation) [27].

### 4.1. Limitations

Similar to most research studies, there was a limitation in the current study. The study was based on self-reported data. As student-athletes report their own feelings, they may be influenced by recent events or unclear of how they feel at the time. This, in turn could impact the way they answered the questionnaires. This is a correlational study and cannot indicate cause and effect, although the results did find a moderate significant relationship between perceived stress and locus of control.

### 4.2. Implications for Future Research

Future research could investigate factors that contribute to effective coping strategies in competitive sport. Coping with stress has been studied in general psychology literature and in sport psychology using the concept of coping style. Future research could examine if student-athletes who exhibit the avoidance approach to coping have a higher level of perceived stress and tend to have a higher external locus of control. The avoidance coping style is divided into two sub dimensions (behavioral avoidance and cognitive avoidance). The behavioral avoidance coping strategy is characterized by physically turning away from the stressor while the cognitive avoidance coping strategy is characterized by ignoring, discounting, or psychological distancing [28]. By researching this topic and identifying a potential relationship between these variables (perceived stress, locus of control, and coping style), it may help coaches to identify student-athletes who may be at risk of depression and anxiety. This could certainly affect their academic and athletic performance and more importantly affect their mental health and well-being. With the NCAA’s 2016 release of new guidelines on dealing with the mental health of college student-athletes, it is evident that continued research efforts to improve student-athletes’ mental well-being are warranted. As noted in the guidelines, mental health is an issue that remains a top concern for the association′s chief medical officer [29].

## 5. Conclusions

The findings of this study give coaches another factor to consider in improving student-athletes’ overall performance and retention. Stress not only has negative health consequences, but also excessive stress may impede performance athletically and academically. Coaches may do well to possibly screen student-athletes for their locus of control and implement interventions for those who score more toward the external locus of control. Working with student-athletes to build their self-concept and self-esteem might help them to develop a more empowered perspective toward their ability to effect change in their personal lives. Consequently, educating student-athletes on how to cope with their stressors might prevent them from becoming overwhelmed when they feel powerless to change their situations. This is especially important coupled with the fact that college student-athletes are less likely than their non-athlete peers to report issues with depression and anxiety and are reluctant to seek help [2]. Practitioners may use the current study’s findings as part of a proactive approach in managing student-athletes’ stress levels.

## Figures and Tables

**Figure 1 ijerph-16-02823-f001:**
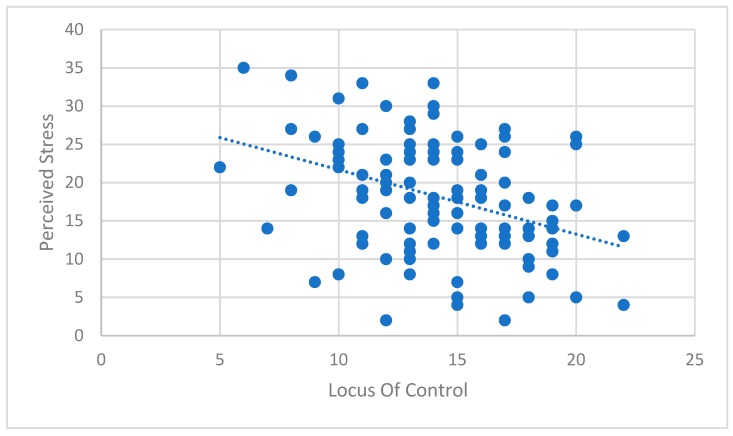
Scatter plot of perceived stress and locus of control.

**Table 1 ijerph-16-02823-t001:** Descriptive statistics.

Variables	Descriptive	Descriptive	Descriptive	Descriptive	Descriptive	Descriptive	Descriptive
	Mean Age	Standard Deviation					
Age (years)	19.69	1.32					
	Mean Score	Standard Deviation					
GPA	3.30	0.50					
Gender	Male	Female					
	28 (22%)	98 (78%)					
Sports Represented	Softball	Tennis	Volleyball	Basketball	Track and Field	Cheerleading	Soccer
	21 (17%)	5 (4%)	12 (10%)	38 (30%)	19 (15%)	12 (9%)	19 (15%)
Academic Scholarship	Yes	No					
	35 (28%)	91 (72%)					
Year in School	First	Second	Third	Fourth	Fifth		
	36 (29%)	37 (30%)	28 (22%)	23 (18%)	2 (<2%)		

*n* = 126. GPA = grade point average; SD = standard deviation.

**Table 2 ijerph-16-02823-t002:** Correlations.

	Gender	Athletic Scholarship	GPA	Perceived Stress	Locus of Control
Gender	1	0.068	0.116	0.323 **	−0.113
Athletic Scholarship	−0.068	1	−0.044	0.015	0.203 *
GPA	0.116	−0.044	1	−0.213 *	0.157
Perceived Stress	0.323 *	0.015	−0.213 *	1	−0.393 **
Locus of Control	−0.113	0.203 *	0.157	−0.393 **	1

** Significant at 0.01 level; * Significant at 0.05 level.

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
