# Peer review of "Sport Locus of Control and Perceived Stress among College Student-Athletes"

_ijerph, 2019, doi:10.3390/ijerph16162823_

Round 1

Reviewer 1 Report

I enjoyed reading your work. I have some minor suggestions. Was self-efficacy assessed? Ln 10 – Please revise your purpose statement in the abstract to better reflect the purpose in the introduction. Ln 117 – The two tables included in the manuscript aren’t really that informative. Perhaps a table with r and p values would be appropriate in the results. Please consider revision Ln 126 -Were any other statistical tests considered? Multiple regression? Ln 135 – Why were the current findings “expected?” Can these findings be juxtaposed to previous findings for more integration into the existing literature?

Author Response

Please see below on how your comments were addressed:

1. Self-efficacy was not addressed in the paper because it was not measured in the study. 

2. The purpose statement in the abstract was updated.  

3. The two tables were removed and a new Table for the demographic information was inserted (Table 1). 

4. Also, a correlations Table was included (Table 2). 

5. We conducted a multiple regression analysis of the variables but the interaction was not significant and we did not add it to the paper. 

6. The conclusion was updated and several references were added concerning why the authors felt the findings were expected.

Reviewer 2 Report

1) Please further interpret r value relating to strength of correlations, results seem misleading stating a "relationship" exists, when it is a low-moderate strength "relationship."  State the findings exactly as is.  

2) Further interpret the difference between scholarship athletes compared to non-scholarship athletes, as I believe differences in perception of situation may exist.  

Author Response

Below is how your comments were addressed in the paper.

1. A statement was provided that the rationale was a moderate relationship. 

2. The Table and results were updated. 

Reviewer 3 Report

Comments:

The article is relevant and appropriate for this journal. Nevertheless, I have several comments:

Lines 79-80: the self-determination theory and the hierarchical model of intrinsic and extrinsic motivation need to be described in introduction.

Authors should precise what is new in this article compared to the previous one in this field. It should appear more clearly.

How do the authors confirm that 126 participants are sufficient to be conclusive?

Line 105: the procedure needs to be more described because it seems to be important to determine the environmental conditions during the surveys. For instance, why coaches could attend to the surveys? It might influence results or not?

Is there a mistake in table 1? From the sentence “Eighty-eight percent of the student-athletes were on athletic scholarship and the mean GPA in the sample was 3.30/4.0 scale. Â», where is the difference with table 1? It is not clear for me.

Why is there no graph on the Perceived Stress Scale (PSS) and the Sport Locus of Control (SIES)? It is the main result.

Minor revision

Please correct: “is defined as defined as a Â» in introductio

Author Response

Please see below how your comments were addressed:

Lines 79-80: the self-determination theory and the hierarchical model of intrinsic and extrinsic motivation need to be described in introduction.

ANS: The paragraph and the title of this section was updated. All changes are highlighted in yellow.

How do the authors confirm that 126 participants are sufficient to be conclusive?

ANS: We surveyed all the athletes we had access to at each institution. 

Line 105: the procedure needs to be more described because it seems to be important to determine the environmental conditions during the surveys. For instance, why coaches could attend to the surveys? It might influence results or not?

ANS: The procedure section was updated and changes are highlighted in yellow.

Is there a mistake in table 1? From the sentence “Eighty-eight percent of the student-athletes were on athletic scholarship and the mean GPA in the sample was 3.30/4.0 scale. Â», where is the difference with table 1? It is not clear for me.

ANS: The table and results were updated. 

Why is there no graph on the Perceived Stress Scale (PSS) and the Sport Locus of Control (SIES)? It is the main result.

ANS: A Figure with this information was added (Figure 1). 

Round 2

Reviewer 3 Report

The authors answered to the requests.